# The Seneb’s Enigma: Impact of a Hybrid Personal and Social Responsibility and Gamification Model-Based Practice on Motivation and Healthy Habits in Physical Education

**DOI:** 10.3390/ijerph18073476

**Published:** 2021-03-27

**Authors:** David Melero-Canas, David Manzano-Sánchez, Daniel Navarro-Ardoy, Vicente Morales-Baños, Alfonso Valero-Valenzuela

**Affiliations:** 1Department of Physical Activity and Sport, CEI Campus Mare Nostrum, University of Murcia, 30720 Santiago de la Ribera, Spain; David.canas@murciaeduca.es (D.M.-C.); avalero@um.es (A.V.-V.); 2Department of Physical Education and Sports, School of Sport Sciences, University of Granada, 18071 Granada, Spain; Dnardoy@gmail.com

**Keywords:** physical fitness, physical activity, sedentary time, self-determination theory, afterschool period

## Abstract

Increasing physical activity (PA) and personal and social values are two of the greatest demands in the current educational system. This study examined the effects of a program based on the hybridization of the Personal and Social Responsibility Model and gamification. A total of 58 students (13.89 years old, SD = 1.14) in two groups (experimental and control) participated during a complete academic year. Motivation, physical activity and sedentary behavior were assessed through questionnaires. Physical fitness was evaluated using previously validated field tests. The results showed significant differences over time between the experimental group (EG) and control group (CG) in afterschool physical activity (APA) during the weekend (*p* = 0.003), sedentary time (*p* = 0.04) and speed–agility (*p* = 0.04). There were no significant differences in motivation. In reference to the intervention, the speed–agility (*p* = 0.000), strength (*p* = 0.000), agility (*p* = 0.000), cardiorespiratory fitness (*p* = 0.001), APA–weekend (*p* = 0.000), APA–week (*p* = 0.000) and sedentary time (*p* = 0.000) increased significantly in the EG. The speed–agility (*p* = 0.000), APA–weekend (*p* = 0.03) and sedentary time (*p* = 0.008) increased in the CG. The implementation of a program based on the hybridization of pedagogical models can be useful in producing improvements in physical fitness, physical activity and sedentary behaviors.

## 1. Introduction

Increasing the physical activity (PA) and, thus, physical fitness (PF) levels among adolescents continues to be a world priority due to the positive physical and mental health benefits associated with maintaining an active lifestyle [1]. Experts stress that PA habits need to be developed early in life [2] and highlight that positive and early PA experiences increase the PA likelihood [3] and reduce sedentary behavior and many risk factors linked to cardiovascular disease [4,5].

Educational centers, particularly in physical education (PE) classes [2,6] and afterschool periods [7,8], have been named as the most influential places for promoting PA and health. In this sense, it is important to determine whether children perceive PE as a valuable, enjoyable and rewarding experience or as a worthless, boring and humiliating one [9].

A recent study highlighted that motivation is the leading theme within the field of sport and exercise psychology across different contexts, including sports, exercise, health psychology and school PE [10]. Motivation is crucial to engaging students in activities from which they can benefit physiologically and psychologically [11]. The self-determination theory (SDT) approach to motivation can be particularly helpful in this sense, as it shows the important role of different motivational types on the cognitive, behavioral and affective outcomes [12,13,14].

Gamification techniques or the art of employing games in the classroom [15] are a complement to increase the extrinsic motivation. Although games provide freedom in learning, where failure allows learning without fear, thus increasing student engagement [16], the use of games is being questioned [17,18,19,20]. In a review study, Hamari, Koivisto and Sarsa (2014) [18] concluded that gamification studies are limited by methodological problems such as small sample sizes, a lack of comparison groups (gamified and non-gamified experiences), short treatments, singular assessments or a lack of validated measures. Additionally, Koivisto and Hamari [21] showed that the appeal of a gamified system might be due to a novelty effect, which could yield decreasing engagement and interest over time. Many gamification elements, particularly the use of badge and reward systems [22,23,24], leader boards and social comparison [25] or competition [26,27,28] might have detrimental effects over the long term for student motivations, satisfaction, enjoyment and engagement, although they are usually used in gamified experiences [29].

Such strategies are thus emerging as an ideal foundation on which to build the teaching of PE. Accordingly, the scientific field has investigated the best way to acquire and develop educational content using different pedagogical models [30], also known as model-based practice (MsBP) [31]. MsBP focuses on the reachable goal of helping practitioners implement methods in a contextualized and confident way [32], modifying the peculiarities of instructional models [33], a type of teaching advocated today by many scholars [34]. In fact, they are called to be the great white hope for teaching in PE due to its broad-ranging and diverse context [32]. The most implemented in educational context have been cooperative learning, sport education, games-centered approaches and teaching games for understanding or teaching for personal and social responsibility (TPSR). For instance, the latter [35] is an MsBP focused on facilitating life skills through several educational tools and strategies.

Despite the common features shared by MsBP, the need to overcome some limitations and the belief that there is not a single model capable of fitting all the content areas led scholars to combine different educational models [36]. This is the reason why several authors believe in the power of MsBP hybridization beyond the combination of educational elements [36,37] and have emphasized that isolated MsBPs present limitations when implemented, because each model is mainly focused on a specific content area [36]. Such models can also help educators innovate to fit the current educational frameworks and fully reach all their students [38], knowing that only when merging different pedagogical models, the benefits related to health are obtained [36,39]. The combination sports education and games-centered approach has been the most widely implemented, but other hybridizations like cooperative learning or sports education with teaching games for understanding or sports education with TPSR have been studied in educational contexts [36].

Many studies in the scientific literature have adopted SDT strategies in PE interventions [40,41,42,43,44,45,46,47,48], but only two have been hybridized with another pedagogical model [45,47]. This suggests the need for further research on the hybridization of pedagogical models within any of the elements of the self-determination continuum in a large PE intervention, comparing an experimental and control group with proper and validated measurements.

The present study therefore examined the influence of a hybrid TPSR and gamification PE intervention within the framework of the SDT on motivation, physical fitness, PA and sedentary time.

## 2. Materials and Methods

### 2.1. Study Design

The intervention program, implemented in two secondary schools assigned to the control group (CG) or experimental group (EG), took place between September 2018 to June 2019 (9 months) in a group-randomized controlled trial [49]. It was a quasi-experimental study, and both groups were selected by accessibility and convenience. Both groups had exactly the same sociodemographic characteristics based on the town hall information. At the beginning of the intervention, the participants (aged between 13 and 15 years) had to belong to the second or third year of compulsory secondary education in either of the two secondary schools selected. The used contents were chosen by taking into account Spanish educational laws [50]. In the first and last weeks of the intervention, different tests were carried out but not before obtaining the informed consent documents (session recording, confidential data treatment and participation in the study) of the students and their relatives. At the beginning and end of the intervention, the participants had to accomplish the sessions for the tests. Informed consent was requested from the students and their parents.

### 2.2. Exclusion Criteria and Participants

All students from both centers were invited to participate in the study. Some exclusion criteria were students who presented any illness or injury that made it difficult to take the cognitive or physical tests or engage in their PE classes normally or need educational support measures. At the beginning, 72 adolescents started the intervention (Figure 1), with 69 (age: 13.92 ± 1.09 years) who completed the experience (37 boys and 32 girls) from the CG (*n* = 39) and EG (*n* = 30).

The participants who, after starting the study, managed to finish it were 58 (80.55%): 32 in the CG and 26 in the EG. Eleven students (5 boys and 2 girls from the CG and 3 boys and 1 girl from the EG) did not complete the study due to dropping out of the study, with effects related to school absenteeism, absent from the PE classes above 20% (*n* = 2), not answering the questions in the questionnaire or doing it wrong (*n* = 5) and not feeling in adequate physical and mental condition to carry out the tests (*n* = 4).

The program, in both groups (CG and EG), was composed by 110 min (two PE lessons) every week. The EG implemented a program based on the hybridization of the TPSR and gamification strategies, while the CG used traditional learning methods focusing on the teacher’s direction and on the student’s lack of initiative in learning, since the teacher had no training in the use of innovative methodologies.

### 2.3. Instruments

Motivation. The motivation behavior was measured by The Echelle de Motivation en Education (EME). This scale (α = 0.801 and 0.823; pre- and post-test) is made up of seven subscales of four items each assessing the three types of intrinsic motivation (IM) (IM to know, to accomplish things and to experience stimulation); three types of extrinsic motivation (EM) (external, introjected and identified regulation) and amotivation [51]. The reliability of each one, pre- and post-test, was IM to know (α = 0.790 and 0.813), IM to accomplish things (α = 0.866 and 0.925), IM to experience stimulation (α = 0.826 and 0.795), EM external (α = 0.682 and 0.733), EM introjected (α = 0.783 and 0.856), EM identified regulation (α = 0. 769 and 0.751) and amotivation (α = 0.853 and 0.858).

Physical Fitness. Before the study, a training workshop was held for the researchers involved in the project for the standardization, validation and study of the reliability of the measure. This intervention has followed similar steps to those implemented in the European HELENA study (Healthy Lifestyle in Europe by Nutrition in Adolescence: www.helenastudy.com (accessed on 27 March 2021)) [52,53,54]. Furthermore, the PF tests used have already shown an adequate validity and reliability in application in the adolescent population [55,56,57]. The tests were (a) a 20-m shuttle run test was chosen to measure cardiorespiratory fitness, (b) the 4 × 10-m speed–agility test evaluated speed and agility and (c) standing broad jump was used to analyze lower body strength. The agility–-coordination and dynamic balance were measured by the hexagon test [58]. All of them were repeated twice, and the best result was recorded, except for the Course Navette test, which was only performed once.

Lifestyle habits. The time used to practice PA (in school and afterschool) and sedentary behavior, exemplified in the use of video games, mobile phone or television, was observed with the Youth Activity Profile-Spain (YAP-S) questionnaire. Cronbach’s alpha was run to check the reliability in the pre- and pos-tests (α = 0.672 and 0,641). Furthermore, it was calculated for any of the categories: PA in school (α = 0.529 and 0.595), afterschool PA (APA) weekday (α = 0.705 and 0.755), APA–weekend (α = 0.829 and 0.710), APA–week (α = 0.790 and 0.733) and sedentary time (α = 0.522 and 0.394). All of them, except the sedentary time variable, showed good indices, considering values above 0.500 suitable when scales are composed of few items [59,60]. Besides, it was previously proven [61] and executed in more intervention studies [62,63].

### 2.4. Fidelity of the Implementation

Hastie and Casey [64] indicated that researchers should provide: “(a) a rich description of the curricular elements of the teaching unit, (b) detailed validation of program implementation based on models or strategies, and (c) detailed description of the ‘program context’ so that readers acquire an accurate and complete understanding of the research design and results obtained.” Parts (a) and (b) have already been detailed in the previous paragraphs. For a detailed validation of model implementation, the research team tried to videotape all sessions. An external observer filmed and, lastly, analyzed 10 isolated sessions (five sessions per group, 550 min) randomly chosen in order to observe if teachers (CG and EG) implemented the methodology based on TPSR and gamification. The same evaluation instrument was used in both groups. Each was distributed in eleven observation periods of 5 min. The camera was installed in the classroom two sessions prior to the beginning of the study to familiarize students and avoid spontaneous behaviors.

Two experts in the implementation of this innovative strategy checked the methodological reliability of both groups, evaluating the frequency that teachers used this kind of technique, choosing between 0 (absence of element) or 1 (presence of element). The inter-observer and intra-observer reliability concordance were calculated by these experts that were previously trained. The inter-observer reliability was assessed between the new teacher and the expert teacher, guaranteeing an agreement greater than 87%, while the intra-observer reliability was assessed by analyzing two different moments over 7 days, assuring an accordance greater than 93%. The checklist instrument was composed of a tool for assessing responsibility-based education (TARE) [65], with additional categories for a gamified intervention [66]. The categories were (1) Mechanics, grants rewards and provides feedback on the accomplishment of the challenges; (2) Dynamics, introduces a narrative thread into the session and generates curiosity; (3) Components, generates missions, realms (groups), roles/status, badges, rankings and markers; (4) Leadership, allows students to lead or be in charge of a group; (5) Task in group, the activity is carried out in a group, with the participation by all team members; (6) Autonomy, empowers students to meet cooperative challenges; (7) Problem solving, works with problem situations that force the student to seek solutions through inquiry or investigation; (8) Choice and voice granting, allows students to reflect, interact and gives them a voice in decisions that affect the development of the class; (9) Group creation and cohesion, favors the cohesion and creation of groups in the proposed activities; (10) Role in evaluation time, allows students to play a role in assessing learning; (11) Transfer, the possibility of applying the values in class to other contexts in real life and (12) Set expectations, it is explicit to students what is expected of them, as well as the content that will be addressed. Total agreement (TA) was calculated using the formula: number of total agreements (NTA) divided by agreements (A) plus disagreements (D) (TA = NTA/A + D). The average in each element was analyzed to calculate their percentages (%). In Table 1, we can see the differences between groups.

### 2.5. Procedure

#### 2.5.1. TPSR Intervention Program

Consistency in the use of this program followed the session format of Hellison [35] but adapted the five parts into four: (1) Initial greeting: the teacher interacted with the students to create bonds, from an emotional side, with them, (2) Awareness talk to formally open the session and ensure that the participants understand the real aim and purpose of the session and program, (3) Physical activity plan: introduce TPSR into the activities analyzing the strategies in order to generate values through these activities and (4) Group meeting and self-reflection time: near the end of the session, the students are predisposed to express their opinion about what happened, indicating their thumbs up (positive evaluation), neutral (medium) or down (negative evaluation).

#### 2.5.2. Gamification Strategies

This part of the methodology is focused on the code to motivate (RAMP) of Marczewski [67]: Relatedness, or being connected with more people in a community, Autonomy, or the sense of being able to give an opinion without censorship, Mastery, or how your skills increase until you reach absolute control and Purpose, or the significance of each step taken.

It is, however, relevant to highlight that the process of integrating game design principles within varying educational experiences appears challenging, and currently, there are no practical guidelines for how to do so coherently and efficiently [68]. However, the following elements, based on the three categories (dynamics, mechanics and components) mentioned by Werbach and Hunter [66], were included as part of the gamified context: Powerful narrative; challenges; class climate; immediate feedback; badges for achievements (“healthy years”) and a final status (squires, Egyptian “melli” and bearers of Seneb).

#### 2.5.3. Control Group Methodology

The CG teacher, non-innovative expert, used a methodology based on direct instruction whose most representative characteristics were centered on the content development and teacher´s actions, without any interaction with the students, and focused on the results of the tasks and not on the learning progress [33].

Each session was divided into three nonconnected parts: (1) Warm-up: body temperature is increased with simple activities such as joint mobility or stretching by using the methodological guidelines based on reception of the contents, (2) Main part: total count of activities designed to improve the assigned technical objective and (3) Cooldown: following the stretches marked by the teacher [33].

This teacher directed and decided the selection of the contents, the strategies to control and present the tasks, the way of participation and the task advances. Moreover, the beginning and the end of sports practice was without the influence of students that did not have to make decisions.

A detailed and extended description of the methodologies on the study has previously been published [69].

### 2.6. Statistical Analysis

Cronbach’s alpha test was used to analyze the internal consistency of the instrument and its validation and assess reliability, in both the pre- and post-test. Then, an exploratory analysis of the data was carried out, through box whisker diagrams and descriptive measurements, in which it was detected that the results could differ between both genders, so this was taken into account in the inferential analysis carried out.

Shapiro–Wilk’s test for normal distributions and Levene’s test of homogeneity of variance test were run to check some of the parametric statistical assumptions. For Levene’s test (*p* > 0.005) and many of the variables in the Shapiro–Wilk’s test (*p* > 0.005), they were positive, except in the control group for amotivation, external and identified regulation, IM to know and to accomplish things, IM general average (only in the pre-test) PA in school, APA (weekend) (only post-test), APA week (only pre-test) speed/agility (SPD-AGI), agility (only post-test) and cardiorespiratory fitness (CF). In the experimental group, there were also some *p*-value significance, amotivation, external regulation, APA (weekday), APA (weekend) (only pre-test) and strength and agility (only post-test). Therefore, not all the starting hypotheses were fulfilled, and it is not possible to consider that the results are conclusive. The variables obtained from the different tools were analyzed with a MANOVA (Multivariate analysis of variance) of repeated measurements, where we named the intra-subject factor “Time” (with two levels: pre-test and post-test) and we named the inter-subject factor “Group” (with two levels: control and experimental). Due to the possible significant effect, the inter-subject age factor and gender were added as covariates. Additionally, the intervention effect size was calculated using Cohen’s coefficient [70] for small sample sizes [71]. The effect sizes were small (0.2–0.5), medium (0.51–0.8) and large (more than 0.8). Statistical analyses were reached with the Statistical Package for the Social Sciences (IBM SPSS 24.0, Chicago, IL, USA), setting the level of significance at *p* < 0.05.

## 3. Results

Table 2 shows the results of the MANOVA test of repeated measurements at the multivariate level. For the inter-subject analysis, there was not significant differences for the age covariate (Wilks’ lambda = 0.736; F = 0.935; *p* = 0.536) or either group variable (Wilks’ lambda = 0.703; F = 1.100; *p* = 0.388), but the gender was significant (Wilks’ lambda = 0.491; F = 2.698; *p* = 0.007). There were also significant differences in the intra-subject analysis between the time and group interactions (Wilks’ lambda = 0.197; F = 10.557; *p* = 0.000) and time and gender (Wilks’ lambda = 0.547; F = 2.153; *p* = 0.028). The fact that the time factor (Wilks’ lambda = 0.770; F = 0.775; *p* = 0.696) was not significant does not mean that there were no differences between the pre- and post-tests, because there were significant interactions with the group factor. This result indicates the time factor (i.e., between the pre- and post-test) should be taken into account depending on the group.

The univariate level was obtained to analyze which variables showed significant differences, taking into account the significance formerly obtained. For the intra-subject factor, significant differences were found in the time and group interactions for the external motivation with identified regulation (*p* = 0.033), APA–weekend (*p* = 0.000), APA–week (*p* = 0.002), sedentary time (*p* = 0.000), speed–agility (*p* = 0.000), agility (*p* = 0.000) and cardiorespiratory fitness (*p* = 0.002). There were time and gender interactions for agility (*p* = 0.021).

Regarding the inter-subject factor, significant differences were obtained in gender for APA–weekend (*p* = 0.005), APA–week (*p* = 0.041), speed–agility (*p* = 0.000), strength (*p* = 0.000) and cardiorespiratory fitness (*p* = 0.000). Then, the differences were analyzed between the CG and the EG separately for the pre-test and the post-test, given the interactions between the time and group factors for many of the variables. Therefore, Table 3 shows the estimation for the participants in the different variables measured in the pre-test and post-test, using means and standard errors and differentiating by group. Besides, it includes the comparison through the Bonferroni correction.

Amotivation (*p* = 0.001), external motivation with identified regulation (*p* = 0.002), APA–weekend (*p* = 0.005), APA–week (*p* = 0.03), sedentary time (*p* = 0.003) and agility (*p* = 0.03) were significantly different in the pre-test compared to between the groups. However, it is important to highlight that there were significant differences in the post-test for EG with APA–weekend (*p* = 0.003), sedentary time (*p* = 0.04) and speed–agility (*p* = 0.04). Comparing the effects of the intervention by observing the results the between pre-test and the post-test for each group, it can be seen that, for the CG, there were only significant differences in speed–agility (*p* = 0.000), APA–weekend (*p* = 0.03) and sedentary time (*p* = 0.008), but the last two variables had a lower value than the pre-test. The results for the EG were significantly better in all the physical fitness variables: speed–agility (*p* = 0.000), strength (*p* = 0.000), agility (*p* = 0.000) and cardiorespiratory fitness (*p* = 0.001) and, moreover, in APA–weekend (*p* = 0.000), APA–week (*p* = 0.000) and sedentary time (*p* = 0.000).

## 4. Discussion

The relevant aim of the intervention was to clarify whether a PE program based on the hybridization of TPSR and gamification could increase the motivation, PF and PA and reduce the state of amotivation and sedentary time.

There has been no study based on the hybridization of pedagogical models that evaluated the variables directly related to motivation. Notwithstanding, only two hybrid experiences [45,47] found improvements in any of the three basic psychological needs (BPNs) that affect motivation. Menéndez-Santurio et al. [45] found significant positive improvements in a student’s relatedness with an intervention based on sports education and TPSR. Gil-Arias et al. [47] studied the impact of hybrid teaching games for understanding and sports education, reporting significantly higher mean scores in praise for perceived autonomous behavior and enjoyment in the experimental groups. Other studies have followed SDT strategies in PE interventions using an isolated MsBP [40,41,42,43,44,46,48]. Burgueño et al. [46] examined the influence of sports education on BPN satisfaction at a basketball intervention and showed an improvement for autonomy, competence and relatedness need satisfaction. Furthermore, Shannon et al. [48], using an intervention based on the Healthy Choices Program, demonstrated improvements in moderate-to-vigorous PA partially by increasing children’s perceptions of the support for autonomy, BPN satisfaction and intrinsic motivation.

Contrary to the conclusions of these studies, the results found in the current study indicated that the use of a hybrid program did not contribute significantly to either the enhancement of variables such as intrinsic motivation or extrinsic motivation or to the reduction of the state of amotivation. Other studies [19,72,73] and a systematic review [29] whose results were also opposite to ours were obtained in isolated gamified interventions with a significant result in motivation. However, none of them measured motivation through SDT or with a validated questionnaire but, rather, with a self-elaborated one [72] or with open questions [73]. Only Férriz-Valero et al. [19] found improvements in extrinsic motivation but not in intrinsic motivation using a validated questionnaire to assess the motivation.

On the other hand, our findings coincide with other studies [17,18,21,23,24] that ratified some gamification elements (i.e., competitive context, badges or leader boards) as harmful to the motivation. According to Deci et al. [23], a combination of a reward (in the form of badges or coins), leader boards and competition affected the intrinsic motivation. Highly motivated students also showed decreased interest over time due to the relative novelty of gamification [18,21].

The data presented in the current intervention exposed that the use of a program based on the hybridization of TPSR and gamification contributes significantly to the improvement of speed–agility and APA–weekend, thus reducing the sedentary time. The EG improved their results starting with significantly worse values than the CG in the pre-test and reached in the post-test the same values in amotivation, identified regulation, APA–week, agility and even higher than the CG in the APA–weekend and sedentary time variables. These data are supported by other study [36], which concluded that only when merging different pedagogical models is when you get benefits in the physical/motor, cognitive, affective and social domains, with respect to the application of only one model. Nevertheless, the TPSR and gamification strategies have never been hybridized. TPSR has only been hybridized to enhance the psychosocial characteristics [45,74,75]. Notwithstanding, the variables related to health, physical–motor and cognitive aspects have never been analyzed through pedagogical hybridizations [36], as in our intervention, which demonstrated that, despite reducing the motor physical involvement in the early stages of applying the model, their levels of speed–agility and APA–weekend were improved, reducing the sedentary time in adolescents. The obtained result, mainly reinforced with PA performed during non-school hours, would be in the same way as the study by Arundell et al. [8], who suggested that this moment of time is an important time for enhancing the PA levels and reducing the sedentary time in childhood and adolescence. In fact, teachers use PA as a means to achieve their goals, regardless of the methodological strategies used. This would be the reason why PA in schools do not have significant differences. It is important to foster this innovative program to the current curricular requirements if we desire to promote the practice of PA in our classes [76] and outside of them [8].

Finally, it is important to consider the important role of the PE teacher, since his/her opinion is crucial to do this kind of intervention properly, influencing the student motivation and reinforcing the initiative and personal effort [77]. In fact, the experiences that students have in the PE area are transcendental for adherence to future sports practices [78], closely related to the conception that the teacher has about it and the way to approach it. The implementation of these methodologies reflects the importance of elements such as social relationships, group work or the student responsibility for homework that entails a higher level of physical activity in the classroom [79]. For this reason, a training formation is implemented to adapt the curriculum according to the educational programs [80] and taking into account that they should be one more component of the program [81] and not the exclusive component, as is usual [82].

In sum, the role of PE is fundamental in any educational intervention program relating the important role of teacher perceptions to the students with more moderate–vigorous practice and cooperative learning [79].

### Strengths and Limitations

There may not be enough representation in this study. Consequently, this study could not be representative if we took into account the equality in the groups. Besides, the post-test was implemented long before the final evaluation period appointed by the educational centers due to massive dropouts. On the other hand, there were some differences in the pre-test conditions between the two groups, and this should be taken into account to have a correct interpretation of the results.

A huge limitation, as has been seen in the discussion, is the nonconsideration of the individual context of choosing whether or not to be included in the gaming experience. It has been proven that a freedom of choice affects the state of motivation in long-term experiences [18]. A limitation that should be taken into consideration was the amount of trained people in the test periods. There were many tests both physical and cognitive, and it could not take too many days to do them. This was one of the reasons why we had to use different people for the pre- and post-tests. Although it may seem like a complication, they were previously taught.

The activity carried out after school could be another variable that could affect the current results. The amount of days per week students received extra PA training should be considered, because this could affect the final scores. It would be interesting to use accelerometers in order to obtain more information about it.

## 5. Conclusions

The results extracted of the intervention suggest that the use of the hybridization between the TPSR and gamification strategies improved the speed–agility variable and the APA on weekends, reducing the time spent on sedentary behavior, but did not improve the motivation variables. Teachers should consider these findings, as potential drawbacks may hamper the outcomes they are trying to cultivate. Although the conclusions could be contradictory, everything indicates that, attending to the scientific literature previously mentioned, there are several variables (competitive context, rewards, badges or leader boards) that could alter the states of motivation in adolescents or that they were even already self-motivated. That would explain the enhancements achieved in terms of the PF, PA and sedentary time.

Decision-makers and administrators for education should consider introducing active and emerging programs to generate a motivational state and, therefore, a healthier and more active lifestyle among adolescents. Such programs should be conscientiously structured depending on the educational context in which they are applied. Future studies involving larger sample sizes could confirm or contrast these preliminary findings.

## Figures and Tables

**Figure 1 ijerph-18-03476-f001:**
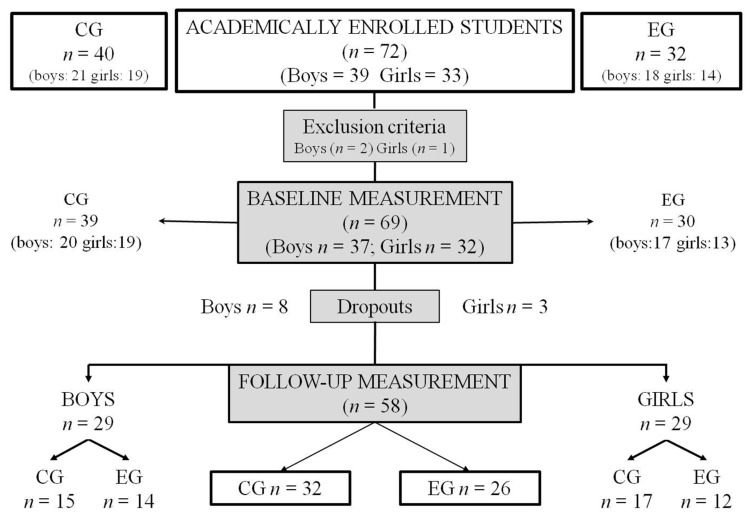
The flowchart of the students. EG: experimental group and CG: control group.

**Table 1 ijerph-18-03476-t001:** Differences between groups in the strategies used. CG: control group and EG: experimental group.

Gamification Strategies	CG (%)	EG (%)
Mechanics	33.33	79.99
Dynamics	0	63.33
Components	0	89.99
Leadership	0	89.99
Task in group	16.66	79.99
Autonomy	33.33	96.66
Problem solving	16.66	66.66
Choice and voice grant	16.66	43.33
Group creation and cohesion	16.66	100
Role in evaluation time	0	53.33
Transfer	0	39.99
Set expectations	33.33	93.33
Global	13.6	74.71

**Table 2 ijerph-18-03476-t002:** Repeated measures MANOVA (Multivariate analysis of variance).

Factor	Wilks’ A	F	*p*-Value	Effect Size
Age	0.736	0.935	0.536	0.264
Group	0.703	1.100	0.388	0.297
Gender	0.491	2.698	0.007 **	0.509
Time	0.770	0.775	0.696	0.230
Time × Group	0.197	10.577	0.000 **	0.803
Time × Gender	0.547	2.153	0.028 *	0.453

Note: * *p* < 0.05 and ** *p* < 0.01.

**Table 3 ijerph-18-03476-t003:** Intervention multivariate analysis (MANOVA).

Variable	Group	Pre-Test		Post-Test		Pre-Post Comparative	
		Mean	SE	Mean	SE	*p*-Value	Dif (SE)
Amotivation	EG	3.21	0.28	2.68	0.32	0.17	0.53 (0.39)
	CG	1.71	0.25	1.93	0.32	0.53	−0.22 (0.34)
	*p*-value + SE	0.001 **	0.41	0.11	0.46		
IM to know	EG	4.43	0.26	4.36	0.28	0.83	0.07 (0.32)
	CG	5.18	0.23	4.96	0.25	0.43	0.22 (0.28)
	*p*-value + SE	0.05	0.33	0.15	0.40		
IM to accomplish things	EG	4.62	0.30	4.46	0.32	0.67	0.16 (0.36)
	CG	5.17	0.27	5.27	0.28	0.76	−0.10 (0.32)
	*p*-value + SE	0.21	0.44	0.08	0.46		
IM to experience stimulation	EG	3.87	0.30	3.67	0.31	0.61	0.2 (0.38)
	CG	4.21	0.27	4.22	0.28	0.98	−0.01 (0.34)
	*p*-value + SE	0.44	0.44	0.23	0.45		
IM general average	EG	4.30	0.26	4.16	0.28	0.65	0.14 (0.31)
	CG	4.85	0.23	4.81	0.24	0.88	0.04 (0.27)
	*p*-value + SE	0.16	0.38	0.11	0.40		
EM external regulation	EG	5.52	0.22	5.94	0.22	0.18	−0.42 (0.31)
	CG	6.17	0.20	5.93	0.19	0.38	0.24 (0.27)
	*p*-value + SE	0.05	0.32	0.96	0.31		
EM introjected regulation	EG	4.79	0.28	4.58	0.33	0.59	0.21 (0.39)
	CG	5.01	0.25	4.88	0.29	0.69	0.13 (0.34)
	*p*-value + SE	0.58	0.40	0.54	0.48		
EM identified regulation	EG	4.86	0.25	5.40	0.23	0.05	−0.53 (0.27)
	CG	6.02	0.22	5.68	0.20	0.17	0.34 (0.24)
	*p*-value + SE	0.002 **	0.36	0.39	0.33		
PA in school	EG	2.93	0.18	3.28	0.18	0.10	−0.35 (0.21)
	CG	2.88	0.16	3.23	0.16	0.06	−0.35 (0.18)
	*p*-value + SE	0.84	0.26	0.85	0.26		
APA (weekday)	EG	2.79	0.18	2.95	0.19	0.40	−0.16 (0.19)
	CG	3.08	0.16	3.19	0.17	0.54	−0.11 (0.17)
	*p*-value + SE	0.27	0.26	0.42	0.28		
APA (weekend)	EG	2.13	0.19	3.34	0.17	0.000 **	−1.21 (0.20)
	CG	2.95	0.17	2.57	0.15	0.03 *	0.38 (0.17)
	*p*-value + SE	0.005 *	0.28	0.003 **	0.24		
APA (week)	EG	2.53	0.15	3.11	0.16	0.000 **	−0.58 (0.14)
	CG	3.03	0.14	2.94	0.14	0.47	0.09 (0.12)
	*p*-value + SE	0.03 *	0.23	0.47	0.23		
ST	EG	2.94	0.11	2.38	0.11	0.000 **	0.56 (0.12)
	CG	2.43	0.10	2.74	0.10	0.008 **	−0.31 (0.11)
	*p*-value + SE	0.003 **	0.16	0.04 *	0.16		
SPD-AGI	EG	13.24	0.22	11.65	0.22	0.000 **	1.59 (0.12)
	CG	12.80	0.19	12.30	0.19	0.000 **	0.50 (0.11)
	*p*-value + SE	0.18	0.32	0.04 *	0.31		
Strength	EG	1.55	0.06	1.68	0.07	0.000 **	−0.13 (0.03)
	CG	1.57	0.05	1.61	0.06	0.17	−0.04 (0.03)
	*p*-value + SE	0.83	0.09	0.51	0.10		
Agility	EG	15.32	0.49	12.58	0.53	0.000 **	2.74 (0.27)
	CG	13.78	0.43	13.43	0.47	0.16	0.35 (0.24)
	*p*-value + SE	0.03 *	0.71	0.27	0.77		
CF	EG	4.14	0.40	5.06	0.40	0.001 **	−0.92 (0.27)
	CG	4.98	0.36	4.59	0.35	0.11	0.39 (0.24)
	*p*-value + SE	0.15	0.58	0.42	0.57		

Note: * *p* < 0.05 and ** *p* < 0.01. IM = Intrinsic Motivation, EM = Extrinsic Motivation, SE = standard error, PA = physical activity, APA = afterschool physical activity, BMI = body mass index, ST = sedentary time, CF = cardiorespiratory fitness and SPD–AGI: speed/agility.

## Data Availability

Publicly available datasets were analyzed in this study. This data can be found here: https://osf.io/bse6h/?view_only=21d14f65235f4aaa8dfd048d8ea55a22 (accessed on 27 March 2021).

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
