# Peer review of "The Seneb’s Enigma: Impact of a Hybrid Personal and Social Responsibility and Gamification Model-Based Practice on Motivation and Healthy Habits in Physical Education"

_ijerph, 2021, doi:10.3390/ijerph18073476_

Round 1

Reviewer 1 Report

The paper entitled “The Seneb´s Enigma: Impact of a hybrid personal and social responsibility and gamification model-based practice on motivation and healthy habits in Physical Education.” reports about the effect of a program based on the hybridization of Personal and Social Responsibility Model and Gamification. The paper is interesting, however some concerns need to be addressed:

MAJOR:

  • A check of the English is recommended to improve the readability of the manuscript.
  • Concerning Table 1, it is not clear to me the results reported. Firstly, it is better to move the table in the Results section, since it reports results. Secondly, it is not clear the p of the last column to what is referred (please better specify in the caption). If it is the p-value of a t-test between EG and CG, it seems that the EG and CG have different initial conditions for several metrics considered in the study. Do the Authors think that this difference could have affected the results? In the results section, the Authors stated that although differences are present in the pre-test condition between the two groups, the variables change differently when comparing pre-test and post-test within the same group. However, how the Authors could exclude that this different changes in the two groups are not related to the different initial conditions? This aspect is crucial, since the novelty of the paper with respect to the literature relies on the comparison of two different methods. Please better justify it in the Discussion section.
  • The Authors use a Repeated measures MANOVA but they do not specify if the data distribution is normal. Please specify if the assumptions needed to employ MANOVA are fulfilled. If they are not, please use a non-parametric test.

MINORS:

  • Line 43, please change “It” with “it”
  • Line 141, caption of table 1, please change “participations” with “participants”
  • Lines 169-188: this part is quite difficult to read, maybe it could be inserted in a table, in order to make the information clearer.
  • Table 2. It is not clear to me to which variables the results are referred. Are the results corrected for multiple comparisons?
  • Table 3. Please use the acronym EG and CG instead of “Experimental” and “Control”.
  • Line 272, please change “Bonferroni correlation” with “Bonferroni correction”.

Author Response

The paper entitled “The Seneb´s Enigma: Impact of a hybrid personal and social responsibility and gamification model-based practice on motivation and healthy habits in Physical Education.” reports about the effect of a program based on the hybridization of Personal and Social Responsibility Model and Gamification. The paper is interesting, however some concerns need to be addressed:

Thanks you

MAJOR:

  • A check of the English is recommended to improve the readability of the manuscript.
  • Concerning Table 1, it is not clear to me the results reported. Firstly, it is better to move the table in the Results section, since it reports results. Secondly, it is not clear the p of the last column to what is referred (please better specify in the caption). If it is the p-value of a t-test between EG and CG, it seems that the EG and CG have different initial conditions for several metrics considered in the study. Do the Authors think that this difference could have affected the results? In the results section, the Authors stated that although differences are present in the pre-test condition between the two groups, the variables change differently when comparing pre-test and post-test within the same group. However, how the Authors could exclude that this different changes in the two groups are not related to the different initial conditions? This aspect is crucial, since the novelty of the paper with respect to the literature relies on the comparison of two different methods. Please better justify it in the Discussion section.

After thinking very carefully this comment, we believe Table 1 does not give any relevant data, and all of them are included in the Table named “Intervention multivariate analysis (MANOVA)”. So, we have deleted this table in order to avoid providing information that confuses the reader.

In the fifth paragraph of the Discussion section, we have added a new sentence interpretating the pre-test differences between the groups and justifying why there has been an improvement in some variables with the program. Finally, inside the Strengths and limitations section, a new sentence has been added warning the risks of misunderstanding the results because of the pre-test differences and pointing out a worse value for agility in the EG.

  • The Authors use a Repeated measures MANOVA but they do not specify if the data distribution is normal. Please specify if the assumptions needed to employ MANOVA are fulfilled. If they are not, please use a non-parametric test.

Some new information about the assumptions to employ a MANOVA has been added in the statistical analysis (Shapiro-Wilk’s and Levene’s tests).

MINORS:

  • Line 43, please change “It” with “it”

Done.

  • Line 141, caption of table 1, please change “participations” with “participants”

Done.

  • Lines 169-188: this part is quite difficult to read, maybe it could be inserted in a table, in order to make the information clearer.

We have modified the description of the strategies and included a descriptive table with the values (lines 180-193)

  • Table 2. It is not clear to me to which variables the results are referred. Are the results corrected for multiple comparisons?

This table is about the multivariate test including group and time as dependent variable in order to see if there are some differences with the intervention taking into account the gender and age as covariables.

  • Table 3. Please use the acronym EG and CG instead of “Experimental” and “Control”.

Done.

  • Line 272, please change “Bonferroni correlation” with “Bonferroni correction”.

Done.

Reviewer 2 Report

I would like to thank the editors for the opportunity to review this work. I think that the information provided here is very interesting and brings practical value to schools. It also reflects the importance of investigating and exploring new methodologies to increase student motivation. I consider the article suitable for publication. However, I believe it is necessary to work on the corrections proposed below before proceeding to publication. 

INTRODUCTION: 
- Rewrite sentence line 43. 
- I would add a more visual example of the models-based practice. I would put more emphasis on describing them. Then describe a specific one, but later talk about hybrids. Line 73. What hybrids? I would mention them briefly in the introduction, their main elements, to contextualise them a little more. 
- The introduction seems to me to be correct, it has a common line and prepares the reader for what the article is going to deal with. I would revise the recommendations mentioned above, but otherwise, I think it is fine. 

METHODOLOGY:

- Study design: what variables do you take into account to determine that the centres have the same socio-demographic variables?
- Instruments: consider redescription line 126
- Fidelity of the implementation. I like the way this point is worded.

DISCUSSION:

- I would include information on the role of the teacher. The importance of the teacher when implementing this type of methodology, and if the results could have been affected for this reason.
CONCLUSIONS:

- I have nothing to add to this section.

Author Response

INTRODUCTION: 
- Rewrite sentence line 43. 

Done.
- I would add a more visual example of the models-based practice. I would put more emphasis on describing them. Then describe a specific one, but later talk about hybrids. Line 73. What hybrids? I would mention them briefly in the introduction, their main elements, to contextualise them a little more. 

Done. Lines 71-76 and 79-81.

- The introduction seems to me to be correct, it has a common line and prepares the reader for what the article is going to deal with. I would revise the recommendations mentioned above, but otherwise, I think it is fine. 

Thank you for your considerations.

METHODOLOGY:

- Study design: what variables do you take into account to determine that the centres have the same socio-demographic variables?

We determined the socio-demographic variables taking into account their educational project in both educational centres. It is an institutional document based on town hall information. Added in line 106.

- Instruments: consider redescription line 126

Done. Lines 143-145

- Fidelity of the implementation. I like the way this point is worded.

Thank you.

DISCUSSION:

- I would include information on the role of the teacher. The importance of the teacher when implementing this type of methodology, and if the results could have been affected for this reason.

It has been included in line 366-373.

Reviewer 3 Report

First of all, I would like to congratulate the authors for a well-constructed, well-founded and well-executed article. for a well-constructed, well-founded and well-executed article.

In the spirit of providing positive considerations to complete the research presented, it would be advisable to the research presented, it would be advisable, however, to increase the references in the theoretical framework of 2021 (there is only one reference) and of 2021 (5 references) in order to make it more up to date.

In relation to the methodology used, there is talk of an experiment with a control group and an experimental group. control group and experimental group, it would be convenient to specify that the extraneous variables have been control
that the extraneous variables have been exhaustively controlled, determine what these variables are and the control that has been exercised on them,
or, if not, to state that a quasi-experiment has been used, which is usual in an educational context. in an educational context, and thus solve this problem.

With respect to the instruments, if they were constructed, it would be convenient to comment on how they were constructed, what process was used to construct them.  how they were constructed, what process was followed, as well as their content validity, normally carried out with expert judgment and pilot testing. and pilot test, and reliability (Cronbach's alpha).
In the case of using commercial tests, these indicators should also be discussed.
Justify why a Studen's t-test was not used to see if the differences in the treatment are significant or the result of chance. significant or are the result of chance. In other words, state that the lambda distribution of wilks lambda distribution is used because it is a multivariate generalization of the F or t distribution. The instruments are the basis of the research and it could be perceived by the reader that there is no clarity in their construction. the construction of the instruments and the items that determine their validity and reliability.

Author Response

First of all, I would like to congratulate the authors for a well-constructed, well-founded and well-executed article.

In the spirit of providing positive considerations to complete the research presented, it would be advisable to the research presented, it would be advisable, however, to increase the references in the theoretical framework of 2021 (there is only one reference) and of 2021 (5 references) in order to make it more up to date.

Some new references from 2020 have been included to make it more up to date.

In relation to the methodology used, there is talk of an experiment with a control group and an experimental group. control group and experimental group, it would be convenient to specify that the extraneous variables have been control
that the extraneous variables have been exhaustively controlled, determine what these variables are and the control that has been exercised on them,
or, if not, to state that a quasi-experiment has been used, which is usual in an educational context. in an educational context, and thus solve this problem.

It has been included in line 105-107. Thanks for your considerations

With respect to the instruments, if they were constructed, it would be convenient to comment on how they were constructed, what process was used to construct them.  how they were constructed, what process was followed, as well as their content validity, normally carried out with expert judgment and pilot testing. and pilot test, and reliability (Cronbach's alpha).
In the case of using commercial tests, these indicators should also be discussed.

New data have been added in Instruments section. Firstly, we have specified the protocol to measure physical fitness. Regarding lifestyle habits variables, we have interpretated the Cronbach´s Alphas of every one of the categories. Finally, we would like to indicate that we used a check list to value the validity of the implementation. And for that, an observer was trained before that to guarantee an agreement greater than 80%. And the inter- and intra-observer reliability concordance was assessed reaching 87% of an inter-observer agreement and 93% for an accordance intra-observer.

Justify why a Studen's t-test was not used to see if the differences in the treatment are significant or the result of chance. significant or are the result of chance. In other words, state that the lambda distribution of wilks lambda distribution is used because it is a multivariate generalization of the F or t distribution. The instruments are the basis of the research and it could be perceived by the reader that there is no clarity in their construction. the construction of the instruments and the items that determine their validity and reliability.

In section 2.5. “Statistical analysis”,we have added some new information about the assumptions to employ a MANOVA in the statistical analysis (Shapiro-Wilk’s and Levene’s tests), justifing why we have used this test.

Round 2

Reviewer 1 Report

I think that the manuscript is improved and it is suitable for publication in the present form. The only change required is in line 289, please change "correlation" with "correction". 

Author Response

It has been changed in line 303, thanks you